# Co-Development of a Tool to Aid the Assessment of Biomass Potential for Sustainable Resource Utilization: An Exploratory Study with Danish and Swedish Municipalities

Andreas Dyreborg Martin 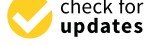

Department of People and Technology (IMT), University of Roskilde (RUC), Universitetsvej 1, 4000 Roskilde, Denmark; andyma@ruc.dk

**Abstract:** In the future, there will be an increased demand for biomass to replace energy and products of fossil origin with renewable alternatives. Such a transition requires action at both the national and local level. Municipalities as key actors need tools to increase the utilization of biomass. One such tool is a means to assess biomass' potential for use. Consequently, this study investigated how a tool to aid the "assessment of biomass potential" (ABP) could support municipalities in Sweden and Denmark to increase biomass utilization. Ten workshops were organized to gain insights into how ABP could be adopted in municipalities. A blueprint of an ABP was developed to aid discussions around four themes: (1) how should the ABP be adopted, (2) which parameters to include, (3) which departments should be involved, and (4) how complex should it be. Many workshop participants saw the biggest benefit of an ABP would be in supporting the municipalities' climate action plans and helping build up the goals and scopes related to biomass utilization. However, for successful adoption of the ABP, many departments need to be involved, which would require building up staff competences. This study shows that ABP could support the increased utilization of biomass.

**Keywords:** bioenergy; bioresource systems analysis; climate plan; local government; local level; participatory method; sustainable transition

## 1. Introduction

With the increasing concerns about the global energy crisis from dwindling fossil fuel resources and the related environmental issues from burning fossil fuels, such as related pollution and global warming from greenhouse gas (GHG) emissions, there is an increased focus on the search for alternative energy options, particularly sustainable renewable-based sources [1]. As one such renewable resource, biomass has great potential to replace fossil fuel with bioenergy and also as a feedstock to produce bio-based products, replacing many of the current fossil-based products. This is already creating an increased demand for lignocellulosic biomasses [2,3]. It has been estimated that by 2050 there will be a demand for 50 Mtoe biomass in the EU to replace 30% of our fossil products with bio-based products [4]. It has also been estimated that bioenergy globally will contribute to between 26 and 35% of primary energy in 2050 [5]. Among the EU28, biomass utilization for bioenergy and renewable energy was 63.1 Mtoe in 2020 [6], but will increase as the demand for biomass for heating, power, and conventional fuels grows [4]. The future mitigation of GHG emissions is therefore dependent on the availability of renewable energy and feedstock sources, including lignocellulosic biomass for biofuels. As a reaction to the new security policy crisis related to the war in Ukraine, both the Swedish and Danish governments are preparing plans to combat resource insecurity and are aiming to speed up efforts to replace fossil fuels with more sustainable alternatives, including biomass [7,8].

It is recognized that to increase biomass utilization, planning needs to focus on the local level, as the local level is where targets from both global and national climate action plans are put into practice [9]. However, the increased number of biomass-based plants arising

from this policy are creating tensions between decision makers and local communities opposed to the construction of new plants [10].

This can affect the municipalities' ability to enable resources to be made available for use in the bioeconomy, as the close relation between the administration and citizens means they need to listen to citizens too [11]. Therefore, planning for biomass utilization needs to recognize lower-level policies regarding the competences of those involved in micromanaging the transition to sustainable solutions. Approaching planning with a bottom-up approach, together with governmental coordination and collaboration at the regional level, could therefore support governments to successfully support access to biomass for energy usage to help them achieve their energy goals [12,13].

Planners at the municipal level currently face a barrier when involved with business stakeholders in the utilization of biomass [12]. On the one hand, enabling biomass' potential is only feasible if certain sales channels are present. On the other hand, bioenergy stakeholders only establish industries if the input materials are present and stable in a certain amount to guarantee supply [14]. This paradox shows the complexity of ensuring a sustainable energy transition at a subnational level. It is at the local and regional governance level that the direct mandate for development is present and where the interests and needs of all stakeholders must be combined with the drive for sustainability [15–17].

To develop a more decentralized structure for the utilization of biomass, the planning for the utilization of biomass needs to aim at gaining local and regional level support [18]. Planning with multiple goals and needs in mind can be complicated and can lead to time-consuming processes to reach agreements and drive actions. It is therefore considered that support tools need to be developed that could, e.g., help shorten the initialization phase [19] and increase the knowledge of which potential biomass resources could be utilized within a geographical area on a regional or municipal level. Many assessments currently focus on biomass potential at a local level [20–22], but only a few focus on holistically assessing biomass potential at a municipal level. Indeed, most biomass potential analysis has a technical or economic scope but tends not to focus on the social and political needs [23]. Another problem with the current assessment of biomass potential is that many studies into biomass potential are not comparable with other local studies. Kautto and Peck [24] assessed five different regional biomass plans and found they covered regional biomass potential, but the resource availability was not comparable trans-regionally [24]. This could potentially lead to less transparency when planning across local communities and decreases the ability to compare local aims and goals.

Another barrier is the gap that exists between the local authorities and the experts who model energy projects, who seem to work in isolation from each other. Amer et al. [25] examined the use of energy modeling and the applicability of the results for planners and practitioners and found that there is a limited overlap between experts carrying out the energy modeling and the energy planning decision makers, which poses the risk that decisions may be purely made based on the technical considerations of external consultants and researchers. In addition to this, policymakers at a municipal level were not consulted in the model development [25]. Similar results were found in a study of 28 municipalities in the Netherlands, where a lack of specific expertise in the municipalities led to a high use of external consultancies [26]. For energy planners and decision makers to be able to participate fully in the energy planning, the models need to be more approachable, as currently municipal energy planners, who are often responsible for a diverse set of tasks, are not specialized to be able to operate the current energy models effectively [26,27]. This points towards the need for models that can be operated with or in collaboration with municipal practitioners, and which could thus be better adapted to locals' needs. Nielsen [16] investigated the success criteria in the initiation stage of biogas plant projects at a national and local level. Many of the drivers for success were related to the local level, where the municipality's ability to facilitate the sustainable transition was a key factor, which could point towards the need for a tool that could aid the "assessment of biomass potential (henceforth, ABP)" and which would include municipality involvement.

To implement biomass models as innovations in municipalities, there needs to be a certain acceptability of such modeling for it to be successfully adopted [28]. Generally, the rate of adoption of an innovation is determined by an innovation's perceived usefulness and can be defined by five criteria [29,30]:

1. Its relative advantage compared to alternative innovations;
2. Its degree of compatibility with current ways of approaching solutions;
3. Its complexity, which should not be perceived to exceed a certain threshold;
4. Its trialability, in terms of the possibilities to experiment with it;
5. That it is offering a certain observability, so the results are visible to others.

The above-mentioned innovation adoption parameters were thus used in the present study to determine how an ABP tool should be formed to support municipalities' decision-making with biomass planning.

This article is based on the findings from 10 workshops held with practitioners within the boundaries of four municipalities located in Denmark and Sweden. The workshops produced insights into how an ABP tool should be formed and how it could be implemented to be useful for municipalities. In the workshop, four main themes were focused on based on the above-mentioned innovation criteria:

1. How should ABP be adopted?
2. Which parameters should be included?
3. Which departments should be involved?
4. How complex should the ABP be to ensure practitioners will use it?

This paper is laid out in the following format. First, we present the theoretical approach in Section 2 based on the participatory method and ABP. Section 3 describes the used methods, covering also the participants, workshops, and the co-developed ABP blueprint. In Section 4, we present the findings from the workshops. In Section 5, we present the discussion and our reflections on how the formation of the ABP should be approached. Finally, we conclude the work in Section 6 and suggest some future work directions.

## 2. Theoretical Approach

### 2.1. ABP

In this study, ABP is defined as an approach to evaluate the bulk quantity of a range of biomasses within a given area and their potential use. ABP differentiates itself from approaches only measuring the flow, as in life cycle assessment, material flow analysis, etc., by enabling a certain interaction with users [31]. This creates the possibility for users to alter the biomass potential regarding addressing certain targeted actions. It is important that the outputs are not a result of the input provided to the user, but that the user can generate their own output to match their own needs [27].

When estimating biomass resources, the assessment can be made in different ways to address different aims. ABP can be calculated with a wide variety of scopes. Many assessments operate with different calculation potentials as restrictions to reflect a certain biomass resource availability. This can greatly affect the amount of biomass available and how it can be utilized in future systems (see Figure 1). This ranges from theoretical potentials, where there are no restrictions within the calculation, to potentials limited by a range of factors, such as the digestible, utilized, and mobilized amounts [19,23,32]. In this regard, calculations of the biomass potential in ABP can be performed to target specific users' needs.

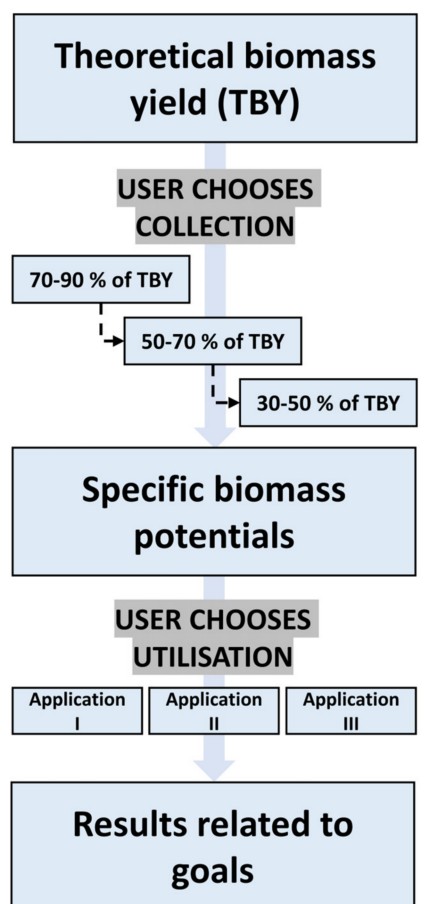

**Figure 1.** Illustration of the ABP process. Figure by author.

The specific biomass potentials as outputs can be utilized in different applications to reach the needs related to the users' goals (e.g., resource recycling and climate gas reduction). Consideration of which biomasses to include in the specific ABP and the level of detail represented in the data can be differentiated, which depends somewhat on the user's ability to understand complex systems [25]. When users interact with the ABP, they will change the definition of the potentials to their needs.

In this study, the ABP will be formed together with the municipalities (users) in workshops (see Section 3). In between each workshop, an ABP was formed according to the users' input at that workshop and presented to the next workshop.

### 2.2. The Participatory Method

In this study, a participatory method (PM) was used as a background method when forming the ABP together with the municipalities (see Figure 2). The PM involves a learning process, in which participants affect the whole process, thus creating co-ownership. All the participants involved contribute to formulating both the problem and the outcome. Repeating the process strengthens the formulation through a process of revision and evaluation [33]. The PM has its roots in the need to involve people in an empowered way, which enables the shift from a top-down to a bottom-up approach [34]. In this study, the PM was used by mapping out diversity to reveal the different sets of information and knowledge held by the participants [35].

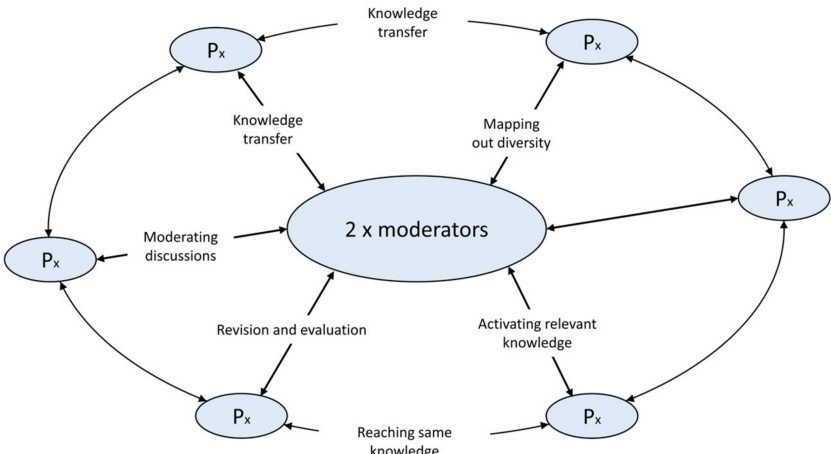

**Figure 2.** Illustration of the PM process. Figure by author.

It is necessary in a PM approach to limit the number of invited participants to ensure a study can meet its time and resource allowances, sometimes even if this means leaving out relevant stakeholders who could otherwise potentially benefit the research. Leaving out certain participants though could also create room for lesser knowledgeable stakeholders, who would otherwise stay more silent and remain in the background of the PM. Aiming to reach the same knowledge level of the participants is therefore relevant to reveal how different information is truly perceived. In this regard, participants who occupy the same position could have different experiences and personalities, which could also lead to a different level of participation. This is also the task of the moderator to encourage every attendee's active participation. The moderator should be aware of the knowledge of the different stakeholders and seek to "activate" relevant information by being empathetic to the different participant types and personalities [33]. The moderator also needs to know the limitations of the discussed surroundings. This is important, because if evaluations and discussions push towards technical or methodical solutions that are not practically possible, the desired outcomes may be unattainable [33]. In this study, two moderators were present to increase the chance of gathering all the relevant information.

## 3. Methods and Data

### 3.1. Research Design

Our approach was an explorative qualitative research design, comprising 10 workshops held with key actors in the subject area within four municipalities. The four municipalities were all part of the InterReg ØKS project GreaterBio and had therefore shown interest in working with biomass utilization. Of the municipalities, three were from the Zealand region and one from the Skåne region (Figure 3). The municipalities are listed in Table 1. For the Bornholm municipality, the initial participants were from Bornholm's Waste Management (BOFA), which solely handles waste biomasses for the Bornholm municipality. The Bornholm municipality has set a goal to be waste free by 2032 and is aiming to recycle all their waste streams [36]. The Odsherred municipality is currently developing a combined climate plan in collaboration with DK2020 [37]. In 2021, the Odsherred municipality established the first Danish combined steam drying and pyrolysis plant. Earlier in 2016, the Odsherred municipality passed a strategic energy plan stating their intention to build a biogas plant and is still attempting to do so these years later [38,39]. Trelleborg municipality's energy plan expires in 2023 and by national law needs to formulate a new one. In the existing energy plan, the main goals are to reduce emissions from transportation by 70% by 2030 (from 2010) and completely replace fossil fuels in the energy system by 2040 [40]. In 2020, the Lejre municipality developed a climate plan to up to 2050, which projects Lejre will be carbon neutral by 2050 [41].

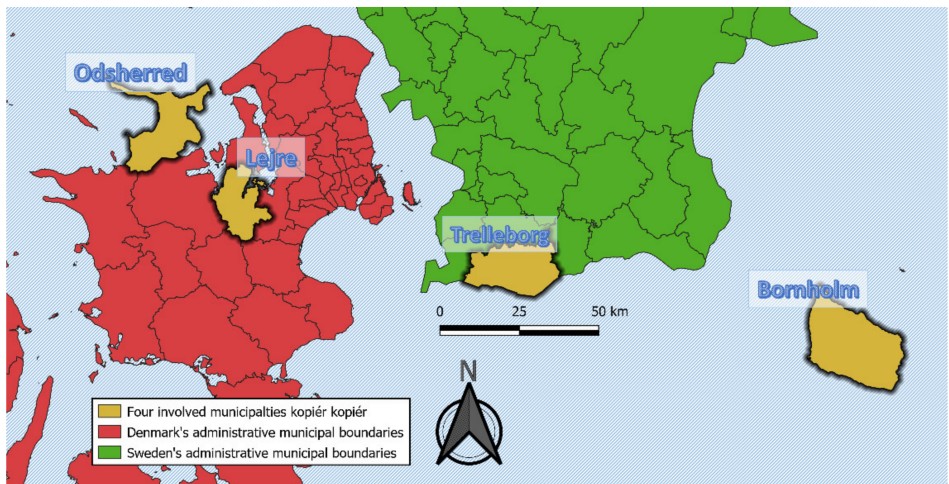

**Figure 3.** Map showing the location of the four municipalities involved in the workshops. Figure by author.

**Table 1.** Employment origin of the participants. Municipality code is used to reference a group of participants from the same municipality in Section 4.

| Municipality | Region | Country | Comments | Municipality Code |
|---|---|---|---|---|
| Odsherred | Zealand | Denmark | Former employee also attended | Od |
| Bornholm | Zealand | Denmark | Businesses invited as well | Bo |
| Lejre | Zealand | Denmark | | Le |
| Trelleborg | Skåne | Sweden | | Tr |

Of the 10 workshops (see Figure 4 and Table 2), the first two workshops (W1 and W2) were held with participants from all municipalities. In W1, a *direction* was discussed about how the ABP should be formed. In W2, a *detailing* process was initiated, where the different parameters and outputs were discussed.

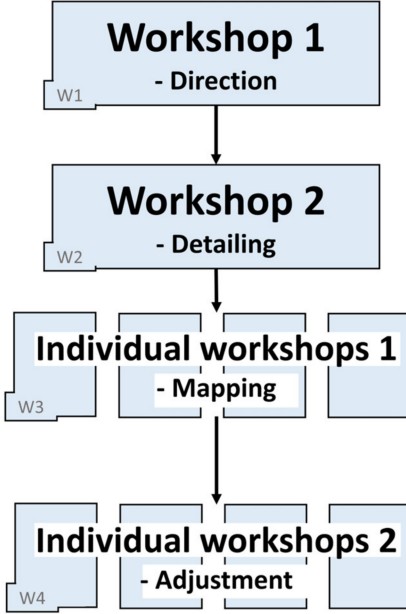

**Figure 4.** Overview of the workshops. W1–W4 are used to reference the workshops described in Section 4. Figure by author.

**Table 2.** Overview of the workshops and research method.

| All Municipalities | | |
|---|---|---|
| Workshop | Focus | Research method |
| W1 | Direction | Group work, discussion about the questionnaire |
| W2 | Detailing | Breakout room discussions, ABP introduction |
| **Individual Municipalities** | | |
| Workshop | Focus | Research method |
| W3 | Mapping | Discussion in MIRO exercises, digital post-its |
| W4 | Adjustment | Testing the ABP, discussion of results, evaluation |

Afterwards, two workshops (W3 and W4) were held individually with participants from each municipality, thus totaling eight workshops. First, W3 focused on the *mapping* process for each individual municipality. Second, W4 used the discussed results and findings from W3 in an *adjustment* process, and the results were presented for each municipality.

In W3 and W4, BOFA (Bornholm) invited both the business community surrounding the municipality and employees of the municipality to the workshops. The other three municipalities only invited participants employed in the municipality. This created different perspectives, which gave the workshops a diverse set of information.

All the workshops lasted between 90 and 140 min. All the workshops were held online on a digital platform, with the participants introduced to the platform MIRO in W3 [42]. MIRO is a visual platform, in which participants can create small boxes of text equal to writing physical post-its. This was used so participants could brainstorm and organize objects in different tasks (explained in Section 3.3). Using a digital platform was beneficial for recording video and audio at the same time, as well as other activities in MIRO.

In the beginning of the workshops, a presentation was made by the moderator, giving an introduction to the process, the daily program, and some general knowledge corresponding to the respective workshop, which helped foster a mutual understanding between the moderator and the participants of what was needed and each person's role in developing the framework [25].

When referring herein to specific workshops, the number of the workshop will be mentioned together with the municipality code (Table 1) or the participant code (Table 2) depending on the statement made in the workshops and by whom.

### 3.2. Participants

Table 3 lists the experience and positions of the 23 participants in the workshops. The experience ranges from 1 to 38 years. All positions counted towards the total experience if participants had had other positions in the same organization or had had the same position in different organizations. The participants employment spanned a broad variety of positions within and related to the municipality, from "Business consultant" positions to broader-scoped positions, such as "Sustainability strategists", and positions referring to specific work tasks, such as "Green areas and bicycle coordinator".

**Table 3.** Experience and current positions of the participants. Participant code is used to reference the participants in Section 4.

| Affiliated Municipality | Current Position * | Experience Related to Position (Years) | Participant Code |
|---|---|---|---|
| Bornholm | Environmental technologist | 2 | B1 |
| Bornholm | International market and business economics | 5 | B2 |
| Bornholm | Plant advisor | 38 | B3 |
| Bornholm | Engineer | 3 | B4 |
| Bornholm | Managing director | 2 | B5 |
| Bornholm | Development consultant | 1 | B6 |
| Lejre | Green areas and bicycle coordinator | 1 | L1 |
| Lejre | Green areas and bicycle coordinator | 5 | L2 |
| Lejre | Waste and recycle employee | 10 | L3 |
| Lejre | Operational planner | 11 | L4 |
| Lejre | Head of road, park, and technical service | 15 | L5 |
| Lejre | Department manager traffic, waste and recycle | 16 | L6 |
| Odsherred | Climate and process coordinator | 1 | O1 |
| Odsherred | Climate and process coordinator | 8 | O2 |
| Odsherred | Business consultant | 5 | O3 |
| Odsherred | Planning employee | 9 | O4 |
| Odsherred | Project manager and climate coordinator | 10 | Q5 |
| Trelleborg | Sustainability strategist | 3 | T1 |
| Trelleborg | Sustainability strategist | 4 | T2 |
| Trelleborg | Operation and maintenance manager, Technical service administration | 14 | T3 |
| Trelleborg | Sustainability strategist | 11 | T4 |
| Trelleborg | Project manager | 17 | T5 |
| Trelleborg | Project manager | 5 | T6 |

* Translated from Danish or Swedish.

### 3.3. Aim of the Workshops

The main goals of the workshops were to understand how the ABP could be adopted in the municipalities. In this regard, an exploratory approach was taken to expand our understanding of this complex topic [43]. The approach taken for the four workshops was to revisit a subject till we could achieve an acceptable saturation of the subject [44–46]. Four themes were explored in the workshops, and all were related to the innovation rate of adoption. As the first theme, ABP adoption was investigated, aiming to understand how an ABP tool could contribute to the municipality. The goal was to discuss the work processes, plans, and protocols within the municipality to help understand where the ABP could fit in with or even simplify current procedures. This could potentially strengthen the integration of the ABP and increase its *compatibility*. The second theme was to scrutinize the relevant departments involvement. The structures of the municipalities as organizations are very different and differentiated in which responsibilities lie within the organization and which responsibilities lie in separate companies. Furthermore, the tasks at hand in each division of the municipality vary and there is a substantial number of different combinations of work tasks and work titles for each employee in the different municipalities. This means that all municipalities are different, and an ABP should be able to relate to the needs of planners with different skill levels and competences. This was supported by the broad array of positions that the workshop participants held (see Table 2). The third theme acknowledged the balance between complexity and the width of usability. It was assumed that more possibilities and features would increase the complexity. Thus, planners would need a higher skill level and more competences. It was therefore a goal to understand how complex the ABP should be to obtain valuable knowledge but without becoming too complex to use. It was thus a balancing act to maintain a high usability for a wide range of employees, while keeping the need for introductory material for the ABP to a minimum.

The fourth theme covered the essential parameters, namely, those around prioritization, goals, and the scopes for biomass utilization. There are many parameters and data elements when mapping biomass potential and the right presentation of the units and calculation foundations should align with the municipalities' goals. This is also closely related to how the implementation should be initiated to support the municipalities' sustainable transition.

One of the key findings from W1 and W2 was that the complexity of the municipality's structures demanded distinct work procedures. This led to the initiation of eight individual workshops, which was the optimum that could accommodate different possibilities of individualization in how to use the ABP. The eight workshops were intended to reveal how individual municipalities would use the ABP to utilize the biomass resources. W3 consisted of different exercises performed in MIRO (see Figure S1) [42]. First, the participants had to link how much influence the municipality had upon different biomasses. Afterwards, the participants had to link the biomasses to current utilization. Finally, the participants had to connect the different actors within the municipalities to the different ways and means they had influence in the use of ABP. The roles were divided into primary roles and different supporting roles, such as knowledge-based and decision-maker roles. Workshop exercises acted as a framework for visualizing the knowledge base, where participants could discuss the needs between departments and actors. The exercises were therefore used to share tacit knowledge between the participants.

Concurrently with W1, W2, and W3, an ABP blueprint was developed incorporating the participants' inputs. In W4, the participants were presented with a finished blueprint of the ABP, where they could comment on the use of the tool and how this would benefit the ABP. In this way, the workshop findings acted as inputs to iterate the ABP blueprint. The last iterations of both the Swedish and Danish ABP blueprints in excel can be provided by corresponding author.

## 4. Results

### 4.1. ABP Adoption

In W1, the participants pointed out several ways of implementing the ABP in the municipality. The consensus was that a wide range of activities within the municipality can affect biomass utilization. This led to discussions about involving the educational system, the business community, board members, and specific planners within the municipality.

The integration of ABPs in the municipally could be difficult: "Implementing biomass tools in the municipality is something that needs to be prioritized to be successful and can take long time to fit in with activities" (W1, T1). For the ABP to fit in with the processes in the municipality, the outputs must in some way speak the same language as the tasks it supports for it to be compatible: "When implementing something new, it must speak the same language as the area of authority" (W1, O3). This is not only to be understood by the specific wording used, but the participants also talked about a general way of thinking: "We write we want sustainability, a circular economy, and $CO_2$ reduction and the whole idea is cascading. The whole idea of using materials where they create the most value. [ . . . ] we work with annual assignments spread over seasons where specific tasks must be done" (W1, O3). Another participant also mentioned the focus on a specific theoretical direction: "They talk a lot about getting the by-products higher up in the [waste] hierarchy [ . . . ], they have an increased focus on by-products" (W1, O2).

One focus of implementing ABPs was integrating results in the municipality's strategic energy plan: "I think it is very important if it can be used as a strategic tool in the climate plan. This is where we could find good use of it. [ . . . ] to make this plan on how to obtain it (the climate plan)" (W1, O2). Another use is in the project initiation phase, where certain parameters need to be mapped: "We are planning a biogas plant, where this is obvious to use. Here biomass resources could be used, such as roadside grass or collected plants from waterbodies" (W1, O2). This was also elaborated by O1 stating that biomass mapping should be part of the new climate plan: "The present goals are in the strategic energy plan [ . . . ], we are in the process of making a combined climate plan in DK2020 [ . . . ]

It is in that that it could be relevant to formulate strategies for biomasses. [ . . . ], so it would be beneficial to use a tool for mapping" (W2, O1). After the initiation phase of a combined climate plan, an ABP could be relevant: "We are currently mapping data to use for $CO_2$ accounting and risk assessment regarding climate adaption. And the next step is to make different work groups [ . . . ]. And this is where this tool perfectly comes into play" (W2, O1).

Participants also discussed using an ABP in connection to the statutory environmental oversights of companies and agriculture. Implementing structures to guide companies could potentially push them in directions towards utilizing their by-products: "Having a checklist, where you can see the by-products and how they can be utilized [ . . . ], sometimes it is also to get it in people's consciousness . . . if there is an economic benefit in doing something" (W1, O2). The participants suggested a checklist for authorities to use at the company visit, which could be used in the process of showing the company's utilization efforts for its by-products, not only for resolving individual utilization issues, but also potentially coupling companies waste streams to one another.

### 4.2. Department Involvement

The four municipalities differ in organizational structure. Under the municipal director, each municipal is organized into 7–11 centers or administrations, which oversee the different policy fields the municipal is responsible for. How these are divided is different in each of the four municipalities. Still, it was mentioned that many competences are sought externally from different organizations, such as consultant agencies, councils, universities, and research centers.

The interviewees also mentioned that the terminology should be identical with the wording used in the municipalities: "It is important to use the same wording as the respective department" (W1, T1).

The participants mentioned that data collection can either be performed by external bodies or by their own team: "We do not have an analytical team. The closest to that is our GIS team, which can do some data extractions [ . . . ] or Getmatic or some things we get from SEGES. [ . . . ] When we must look at the industry, our business centers are helpful. There, we can also find data from them" (W2, O3).

The way the municipality is structured with regard to tendering, sometimes makes it a challenge to obtain data from activities undertaken by private contractors: "The organization of the municipalities is such that many things that happens with collection are laid out in some public tenders, where there is an entrepreneur, who will have an assignment" (W2, O3).

B4 (W3) also explained that in the Bornholm municipality, the department for properties and operations sits with the practical day-to-day management of roads, forest, etc., apart from beach-cast seaweed, but the decision makers in management are situated in another department.

### 4.3. Complexity and Width of Usability

Overall, there was common agreement that when it comes to the parameters and functions, the more details the better. However, when the different municipalities were presented with the structures of a tool in W4, working around the wide range of different biomasses, applications, and parameters presented was, in general, a barrier.

The operational staff present from Le (W4) and Tr (W4) expressed concerns about using an overall ABP. They had a hard time seeing the applicability of the tool in their administrations. Both participants L4 and L5 said directly that using such a tool was not going to happen: "If it must be usable, then we should have roadside grass. Then it should be more adapted to us, so we don't need to take poultry and cattle manure into our consideration" (W4, L5). In this regard, the ABP needs to be directly connected to the specific tasks. The operational administrative personnel therefore seemingly found the ABP too broad and lacked sufficient details that could be used in decision making. At the

same time, the ABP had too many unnecessary calculations that did not seem relevant and that increased the complexity significantly (W4, L4, L5, T3, and T4). On the other hand, environmental planners, who had a broader view, found it important to include all biomasses: "I think it would be nice if we found a solution to all biomasses that are on Bornholm (in the area), which creates challenges" (W3, B1).

It was also argued that the ABP should mainly be aimed at administrative personnel: "It has to be on the boards and clerks' level before it can have an impact in the organization" (W1, T2). Furthermore, it was elaborated that the tool could be implemented through the most skilled staff on the subject before introducing the ABPs to the rest of the municipality: "First it must be us, who know a little bit more about it, that must use it. But it also must be used more across departments [ . . . ] I know that our business consultant would also find it interesting to use such a tool [ . . . ]. They know the companies, and which resources they have" (W1, O2). This also affects the educational and introductory aspects needed to become familiar with the ABP: "I think you have to use a lot of time to get familiar with it [ . . . ] I feel like, you need some kind of instruction" (W4, L2).

The participants also suggested having different levels of complexity in the ABP, which would enable users to increase the complexity themselves: "If you could make different levels in the ABP [ . . . ]. You could begin as broad and simple, where you can get something mapped and how can they be utilized. Then you can dive deeper down. So that you do not meet a wall of a steep learning curve" (W1, O1). "You should start the ABP with a success" (W1, L1).

The participants also discussed the importance of having simple outputs, which is understandable for a political system: "If we are doing something that has to go through a political process, it has to be pitched pretty simple and understandable or else we won't get a mandate to continue with it" (W2, O3).

One participant from the Danish Agriculture and Food Council at Bornholm mentioned that they have their own procedures to adhere to, and do not see any need to engage with new tools: "Honestly, I don't think [we would use ABPs] [ . . . ] we utilize large amounts, and we use tools developed nationally from the Danish Agriculture and Food Council. And we have a lot of planning tools . . . " (W3, B3). This view shows that some actors have a high threshold to overcome to consider ABP as a useful tool.

*4.4. Parameters to Include*

The participants also discussed the different parameters needed in the models, as shown in Figure 5. The participants were not necessarily fully qualified in knowing all the parameters for the different utilizations but expressed these through the various goals the municipality aimed at around biomass utilization, such as making sure that waterbodies are kept clear of plants so water can be drained from agricultural fields or cutting roadside grass, which would reduce snow on roadways (W3, Lr). The goals and parameters were linked together with certain utilizations expressed in various scenarios.

The participants mentioned that $CO_2$ reduction in itself is not always enough as a lone parameter to get the political system to address an issue: "We experienced that we have to couple goals up with something else other than $CO_2$. It's hard to get something through politically if you only can say that it will reduce $CO_2$ by 20,000 tons. For example, with a biogas plant, [it's better] if we [can also] talk about what it does for agriculture, for nutrient uptake, for the reduction a nitrogen to Isefjord, more jobs. How it can benefit businesses, settlements, the environment, and the side effects there are that will have a big impact for the climate agenda [ . . . ] Or can energy savings or other forms of climate actions be aimed to solve other goals?" (W1, O2).

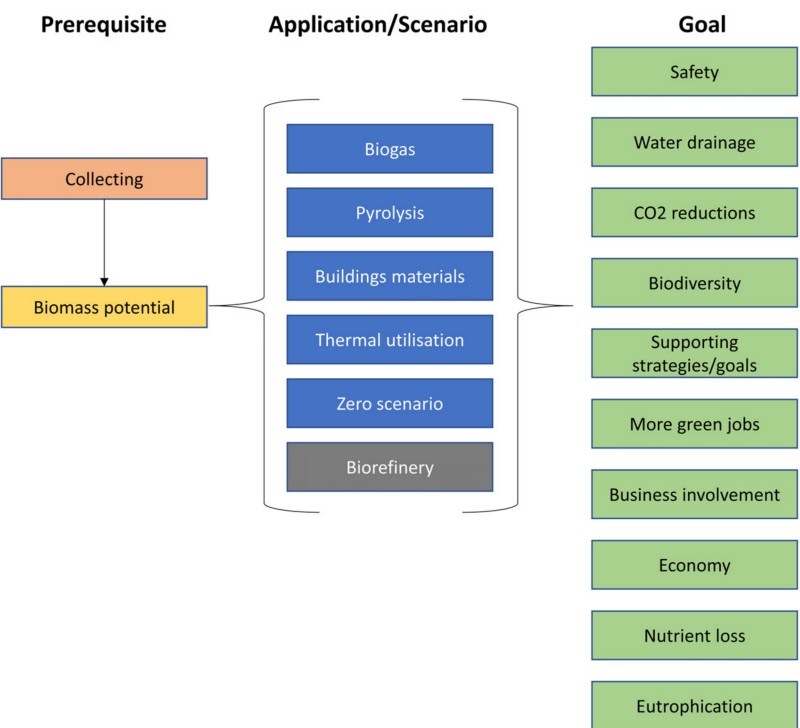

**Figure 5.** Different goals elaborated by participants and the "applications/scenarios" connected to the goals. Prerequisite inputs are the biomass quantities. Figure by author.

The interviewees mentioned that backing data on different biomass aspects, such as economic parameters, are relevant in the planning: "Eel grass could be relevant in Odsherred Biogas Plant that hopefully will soon be established. But there was also the problem that we could only collect 4000 tons yearly because it was a Natura-2000 area (Nature Protection) and is it then feasible to invest in a machine to separate out the sand?" (W1, O2). Furthermore, it was elaborated that the prioritization of the economy is key when working with a low budget: "It is not that the politicians do not understand $CO_2$ reductions, it is because, if you are a rural municipality, you simply cannot afford to not think economically. They have so many on social income and many issues and they are trying to get newcomers in, but their villages are dying out" (W1, O2). Climate solutions must also be holistic and solve other problems, to attract and increase political attention: "We shall close so many schools, we must cover the elderly area [ . . . ] So it is a luxury thing they have a hard time prioritizing. [ . . . ] They really want this. So, if you can show them that there is a way a rural municipality can be a part of such actions, then they will be happy. Because when the citizen says that they are closing schools, they [politicians] can show that they are pushing on something else, for example the increase in newcomers [ . . . ] it is a question of hardcore priorities" (W1, O5). Planners are dependent on political support to go forward with projects (W2, O3).

Another way of designing the ABP was to show the fluxes of biomass from and to the municipality's border: "A model that shows what comes in and what goes out of the municipality [ . . . ] I don't really know what goes out and what comes into the municipality regarding biomass" (W1, L1).

Another topic that was discussed was about the different data needed for specific activities: "It is for me a lot about low-practice issues [ . . . ] It [biomass] has to be collected, where should it be stored and it degrades a lot to begin with; who shall collect it, can you do it while you are there anyway; and working hours is especially [important to consider] when it is running wild" (W1, L1). The participants O3 (W3) and O1 (W2) also mentioned the need for specifying the different types and quantity of biomass and how these can be worked with.

L2 (W2) mentioned the need to solve specific tasks. For example, they need the specific output from haying the municipal's green areas and need to differentiate how broad roadside grass is cut in spring and autumn. On the other hand, B1 (W2) mentioned that they are more interested in the overall fraction, since they are an island, and everything ends up in the same place anyway.

Some quantities are also already known by the municipality, such as garden and park waste: "Quantities of garden and park waste are known at the technical service administration" (W2, T2). O3 (W3) also explained that municipal recycling facilities have a license plate recognition, so they know the quantities processed through such channels.

It is also relevant to distinguish between the ownership of the biomass (W2, L2, and W3, Od, B4). Biomasses such as food waste, plant trimmings, and roadside grass can accrue both public and private ownership, which changes the municipalities' options to influence utilization (W3, Od). This was also elaborated by B4 (W3), stating that the municipality only has direct influence on biomass from public managed areas, such as beach-cast seaweed, forest residues, and grasslands.

The quantity of biomass is not always the main priority. Another parameter is the potential benefits for biodiversity from collecting biomass. At Bornholm, certain nature action plans have been formulated, where the main purpose is not the biomass itself but the management of nature to increase biodiversity and nature's ecological condition (W3, B4).

Another point was also made that if municipalities were to use the same ABPs in their calculations, it would be easier to cooperate and compare: "It is also relevant to use the same mapping method in all municipalities. Then we can to a greater extent compare with each other and collaborate, if we use the same mapping method and do not use different consultant agencies" (W2, O1).

*4.5. Summary*

The results and findings for the four themes (Sections 4.1–4.4) are summarized in Table 4.

**Table 4.** Summary of the findings for the four themes.

| ABP Adoption | Department Involvement | Complexity and Width of Usability | Parameters |
|---|---|---|---|
| Use same wording/terminology as municipal tasks. | Use same wording/terminology as department. | Not usable for operational staff. | Holistic solutions are needed for the ABP to be approved in the political system. |
| Same general way of thinking. | Data collection is sought both internally and externally. | ABP should be aimed environmental planners within the municipality. | The bulk quantity of biomass needs to be combined with its utilization and economic aspects. |
| Integration in climate plans. | Challenge to obtain data from subcontractors. | Creating levels of complexity would benefit the usability. | Biodiversity is an important parameter in land management. |
| Project initiation phase as a mapping tool. | Public tendering needs to focus on data collection as well. | Simple outputs are needed that would be understandable for the political system. | Comparable data and data processes are needed between municipalities. |
| Use in statutory environmental oversight of companies. | Decision makers and practical day-to-day management sit in different departments. | | |
| Long-term implementation is needed. | | | |

*4.6. Blueprint for the ABP*

Concurrent with the workshops, a blueprint of an ABP was developed. The blueprint corresponds to the results and findings revealed in the workshops in the four previous

result subsections. The blueprint focuses on data collection and processing for mapping and for use in climate plans. The ABP was set up so that the user has access to outputs within the first few minutes of opening the program. After that, the user has further possibilities to change different parameters to increase their output possibilities. This acts as a way to incorporate different levels of complexity in the ABP blueprint, which many participants desired. The blueprint focuses on simple outputs that would be usable for environmental planners within the municipality. The ABP was made in excel so all municipalities have easy access to it.

In Supplementary Materials, sources for all the parent data can be accessed as well as both the Danish and Swedish excel blueprints of the ABP (in their respective language) can be provided by the author.

The ABP consists of biomass potential calculations. These potentials can be used separately, but also in the ABP for further calculations related to different applications (see Table 5). In Figure 6, a simplistic framework of the ABP blueprint is shown corresponding to Figure 2. Users can alter the biomass potential and choose a current and future intended application. As an output, this ABP focusses mainly on nutrient flow (resource) and carbon flow (emissions). This enables practitioners to see the output when changing the utilization for specific biomasses. The output can be used to create goals that can be added to climate action plans or used in communication activities and other related efforts.

**Table 5.** Applications chosen for the ABP by participants.

| Application |
| :---: |
| Waste incineration (CHP) |
| Anaerobic digestion (Gas grid/CHP) |
| Fodder |
| Biomass (HOP/CHP) |
| Fertilizer/Soil improvement |
| Not collected |
| Application unknown |
| Composting |
| Pyrolysis (HOP) |
| Bedding |
| Sold |

CHP: combined heat and power. HOP: heat-only plants.

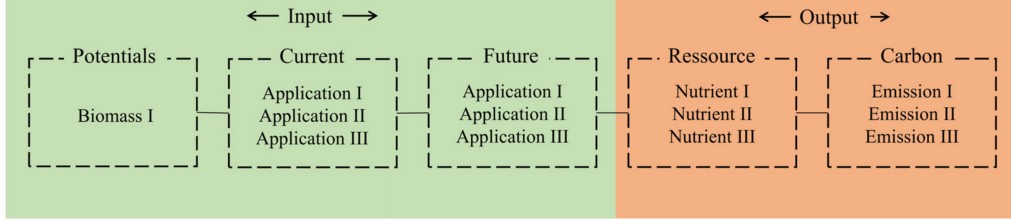

**Figure 6.** Simplistic framework of the ABP blueprint.

Data for the applications were retrieved from the Danish Energy Agency's technology data [47]. The applications and biomass were chosen based on participants' MIRO sketches in W3. Not all biomasses that were mentioned were included. Only open-source data were chosen, so that practitioners have the possibility of sharing results in any detail and format.

## 5. Discussion

### 5.1. Maintenance of the ABP

The complexity of the ABP can be increased providing this does not exceed the comprehension of the target group that uses it. However, another bottleneck is that the framework needs certain maintenance to ensure it remains operational and fit for purpose. How a tool lives on depends on how its monitoring is framed, which should support

periodically updating the data or tool's functions without high cost and depending on the availability of data [48]. This requires that the sources used are accessible by the municipality themselves or that they are somehow reachable. New ways to use technology to gather biogenic data are constantly emerging, e.g., remote sensing in urban areas, hour-to-hour resolution, GIS, crowdsourcing [49–51], but with the increase in possibilities, the discussion arises about which to prioritize and which limitations to incorporate. Many parameters can be relevant, but the ongoing loading of extra details can decrease the participation of municipal planners and their ability to transparently assess the possibilities, pass on simple outputs to the relevant political system, and compare outputs with other municipalities.

### 5.2. Practitioners' Involvement in Adoption

When juggling with a comprehensive array of parameters and quantities, the complexity heightens. However, detailed calculations were mentioned by the participants in the municipality as important to address climate problems, such as mapping, project initiation, and climate plans. On the other hand, if ABPs have too much complexity, it may put off operational staff from using them. Many of the participants understood the concept of the ABP and could navigate and alter the different parameters. When initiating the workshop, the goal was to create a framework for a tool that could be built that both planners and operational staff in the municipality could find useful and would use with the same models. However, the differences in competences between the departments suggested that models should be separated so that operational administration would have tools for specific tasks and the department in charge of the broader environmental planning would have a tool for their specific needs. Still, to be able to implement ABPs, the participants emphasized that there is a need for a certain building up of competences in the process. Therefore, a model cannot stand alone when practitioners within the municipality must operate in the field of biomass utilization. This was also pointed out by Amer et al. [25], who stated that practitioners should be a part of producing outputs for models. Gaarsmand [17] also investigated the implementation of renewable energy plants, where one of the main drivers for success was in establishing competences within the group of involved practitioners. Building up competences, however, takes time, and if changes are not implemented long term, they will likely not be adopted. So, the ABP needs to use specific terminology and wording in line with the language of the municipality, while the municipality also needs to adopt to the ABP.

### 5.3. Holistic View

Another dimension in addressing the holistic view is how value is predicted. The participants juxtaposed local value creation and the involvement of citizens and the business community with more technical parameters, such as $CO_2$ reductions and nutrient recirculation. For instance, implementing energy projects is not only dependent on financial and technical dimensions [17]. This was also reported by Vaidya and Mayer [52], who reported that regional participants valued quantitative and qualitative criteria and indicators equally. So, thinking in these stepping stones in or around ABPs would strengthen the implementation. Therefore, an evaluation process is needed that could bridge the desired outcomes with a broad scope of political incentives. Choosing technologies for biomass utilization to transition to sustainable solutions should therefore not purely be based on calculations. Sustainable solutions are concepts, with certain inherent benefits that can fit specific needs. This ultimately makes it very complex to narrow down how the transition should be planned. However, involving municipal planners will undoubtedly secure knowledge about prioritization in the local political landscape. In building an ABP, the quantifications need to be dynamic to be able to adapt to non-quantifiable outcomes, such as the political landscape, local incentive, and the knowledge base, which can be very different. Defining the level of detail against the thoroughness or ensuring dynamic complexity can be complicated, cf. [31,53], thus making broad ABP products that cannot

be used as standalone elements nor be used for detailed planning. For detailed planning, an individualization process is needed that would add the detail to an otherwise too simplistic picture.

*5.4. Methodological Reflections*

This research has shown a way of producing a blueprint for an ABP concurrent with the participatory involvement of a range of stakeholders. The group of participants was therefore part of the whole process and had the opportunity to influence the development of the ABP, from its creation in the beginning to its evaluation at the end. The author hopes that this approach can potentially reduce the gap in the use of ABPs between experts and practitioners. Still, the tool was not created by practitioners alone. There are many hidden calculations that are not easily understandable for all using the ABP blueprint (e.g., efficiency of combined heat and power plants, carbon content of biomass, standard data for animal husbandry). There will always be the need for a certain simplification to encourage more practitioners to use the ABP.

The workshop participants in the study mentioned that they always have a limited number of resources available [54]. This was also considered when designing the timeframe and number of workshops. Furthermore, by choosing to use a digital platform, participants needed to devote less time on travel and found it easier to schedule. Still, W3 for *Le* ended up with last minute cancellations from some participants due to a lack of time, which resulted in less participants attending the workshop than desired. Maybe this could have been avoided by decreasing the duration of the workshops but setting too little time aside might also have increased the risk of losing or missing valuable information. This was especially true with W3 for *Od*, where all tasks could not be undertaken in the original timeframe, resulting in a follow-up meeting needed. In another case, too little time resulted in the participants feeling like they were swept to the background, which made it hard for the facilitator to unlock their knowledge. Building up of trust is important, especially when striving for radical changes between actors with very different backgrounds [55]. This became apparent in W4 for *Tr*, where Tr needed to hold a follow-up meeting without an outside facilitator and instead sent their meeting minutes to the author.

The participants also came with different competencies and settings of time to discuss the different approaches in various iterations, which helped raise the level of knowledge among the participants [56,57]. This was especially true while using digital boards, such as MIRO, where ideas and thoughts could be easily written down in parallel to the discussion on a display for all participants to see, which made it easier to grab hold of others' ideas and points. Having a display where points were written down also made it easier to get back to earlier *crossroads* if the discussion had gone in a specific direction, thus reducing the risk of losing valuable knowledge.

ABP as an innovation was an important focus in the workshops. Enabling biomass resource use through municipal involvement is an unavoidable step. By understanding how municipalities perceive ABP as an innovation and what incentives lie behind this can help pave the way to a sustainable energy transition. In this study, the five criteria used for the rate of adoption were very useful to understand how an ABP tool could be included in municipalities.

## 6. Conclusions

This article presents valuable insights into how practitioners at the local level could be part of building up an ABP and how an ABP tool could be used in municipalities to help lower the barriers for the utilization of biomass. This study involved a broad range of knowledge transfer and at the same time the study format made it possible for practitioners to comment on and critique complex systems.

Many of the practitioners in the municipality could see a wide array of possibilities in participating in the development of an ABP, but they felt that first the ABP needs to be directed towards a specific area and cannot contain too many purposes. Another remark

was that the ABP cannot stand alone, and biomass utilization needs to be linked to multiple purposes, such as job creation, $CO_2$ emissions, and financial parameters to ensure political support. In fact, there are many parameters that could be included or need to be built up around the ABP for it to be fully beneficial for municipalities.

The participants were in fact able to navigate around an ABP but needed instructions and guidance. This means that there is a possibility to create systems that could improve the inclusion of municipal practitioners in the alteration and improvement of biomass potential, but this would require building up competences in the municipalities, which would be time consuming and needs to be prioritized.

When it comes to the ABP, the practitioners saw many angles in which an ABP could be useful in supporting the municipalities. Three of the four municipalities were in the process of making a climate plan and needed input on biomass utilization. In this process, many ideas arose for the use of an ABP, such as integrating it in certain climate plan work groups, using it as a mapping tool, or to add biomass strategies and goals.

**Supplementary Materials:** The following supporting information can be downloaded at: https://www.mdpi.com/article/10.3390/su15129772/s1.

**Funding:** This research was funded by Interreg ÖKS under grant #: 20203414 (GreaterBio) and #: 20358661 (PowerBio).

**Institutional Review Board Statement:** Not applicable.

**Informed Consent Statement:** Informed consent was obtained from all subjects involved in the study.

**Data Availability Statement:** All data are available upon request.

**Conflicts of Interest:** The authors declare no conflict of interest.

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
