# Peer review of "Co-Development of a Tool to Aid the Assessment of Biomass Potential for Sustainable Resource Utilization: An Exploratory Study with Danish and Swedish Municipalities"

_sustainability, doi:10.3390/su15129772_

Round 1

Reviewer 1 Report

Decision: Major Revision

Summary

I think that it is an important issue to develop a tool to evaluate the potential of biomass utilization and how these tools could be supported local governments, which is the subject of this study. In particular, I agree that the cooperation of local governments is indispensable in establishing policies for renewable energies such as biomass.

However, the following problems in this study make it difficult for the reader to understand this study.

ž   The vision of the two Theoretical Approach” described in 2. is not clearly presented.

ž   The correspondence between the methodology and the results is not clearly shown. Then it is difficult to understand what was obtained as results from this study method.

ž   I have the impression that the result parts are only lists of opinions that came out of the workshops. Therefore, the manuscript is redundant and is difficult understood for readers what was obtained from this study.

As such, I think that this manuscript should be improved. Please revise your manuscript to easier understood for readers.

Each part

2. Theoretical approach

You applied two theoretical visions “ABP” and “The Participatory method”. However, I difficult understand these visions from your explanation sentences. Please clearer explanation and should add explanation figure for these approach images.

2.1 ABP

You wrote “ABP is defined as an approach in which to evaluate bulk quantity of a range of biomass within a given area”. Please add detailed explanation estimation method of ABP and figure ABP estimation result of showing stakeholders at methodology part or result part.

Figure.1

Please add map color legends. Please add scale and orientation.

Figure.2

Please use the “same word” in this figure and in the your manuscript, e.g., "W1".

Please add a table with a specific description of the research method corresponding to Figure.2.

Line 194-210

Is this method like the Japanese Kawakita-Jiro method?

Please explain more clearly what you could get from this method and what you heard to stakeholders, as it is an important part of your research methodology.

4.Result

This section seems redundant as it very detailed describes what was said each stakeholder at the workshop. In addition, it is difficult to understand what is important from this part explanations. Please summarize what was said stakeholders at the workshop and change the manuscript that is easier for the reader to understand.

There are four levels of workshop hierarchy in this study, “W1”-“W4”. Please interpretation of this research results by each hierarchy level workshop shown in the research methodology or what happened at each hierarchy level workshop in the four targeted regions.

Figure.5 & 4.5 Blueprint of ABP

Please clearly describe method in the “Methodology part” how you obtained these results.

Discussion

Please revise the manuscript to show the correspondence between the results obtained in this study and each subsection at this part.

Line 31

”Biomass is of great importance” ->maybe ”Biomass is great importance”

Please check your manuscript once more.

Reviewer 2 Report

The article describes the biomass potential for its sustainable utilization concerning municipalities in Denmark and Sweden. The topic is very important esspecially that these countries have quite a big potential of wood production in a large forestry areas in each country. The involvment of the local communities to take care of mentioned problem is very practical and useful.

Please devide Section Conlusions on separate Subsections to present the most important findings of the study, what would be very valuable.

English language is aceptable. Please make some improvements in Section Conclusions.

Reviewer 3 Report

Thank you very much for the invitation to review this article entitled "Co-development of assessment of biomass potentials for sustainable biomass utilization: An exploratory study with Danish and Swedish municipalities."

The article presents a current and interesting topic that falls within the scope of the journal and should be considered for publication if it meets the necessary requirements for a scientific article.

The article is well constructed and presents relevant information, demonstrating that the author is knowledgeable about the process addressed in this work.

In my opinion, the article could be published after correcting some issues that could improve the final quality of the work.

Recommendations:

Title: The author should choose another title to avoid repeating words, such as "biomass."

Abstract: The abstract should be simplified to a limit of 200 words.

Keywords: Keywords that do not appear in the title should be chosen. "Biomass potential" and "resource potential" should be merged into one keyword.

References: The author should cite additional, more recent references that address these topics. The formatting used by the MDPI journals should be utilized.

Line 37 - The decimal separator notation should not be "," but ".". The entire document should be reviewed.

Line 89 - Colon is missing.

Section 3 - Text should not follow the main section when a subsection follows. This text should be removed or transferred to another location where it makes sense. If necessary, a "Framework" subsection can be created. The entire document should be reviewed.

Line 180 - Table 1. If "." is used after "1," there is no need for "-". The entire document should be reviewed.

Figure 1 - This figure is of poor quality. It is possible to create this type of image using the free software QGIS and using spatial data provided by the geographical services of countries. The scale, relative location, cardinal orientation of the map, and legend are missing.

Figure 2 - This figure is of poor quality. It can be created using Lucidchart, which allows for professional-looking images for articles.

Table 2 - If the full width of the page is used, no words will be cut off. In any case, hyphenation should be activated.

Figure 3 - This figure appears to be irrelevant since the information must be in continuous text.

Figure 5 - Poor quality. See the recommendation for Figure 2.

Line 479 - Recommendation the same as for line 180.

Line 590 - Funding reference is missing.

Moderate corrections are need.

Reviewer 4 Report

The draft is well prepared and need minor revisions with respect to inclusions of relevant citations and moderation of English language. After the revisions, the manuscript can be accepted for publication.

The English language can be moderated and rewritten for enabling maximum researchers across globe to understand the scientific objective.

Round 2

Reviewer 1 Report

I think that the added parts and revisions make the main idea of the paper easier to understand for readers.

This area is not easy to understand for readers who are not familiar with this field, so I hope you could check to confirm that the message is easily understood.

There was a misspelling in the manuscript, which should be corrected.

This paper have some of missspelling. Plese correct these parts.
